# “What If I Die and No One Notices?” A Qualitative Study Exploring How Living Alone and in Poverty Impacts the Health and Well-Being of Older People in Hong Kong

**DOI:** 10.3390/ijerph192315856

**Published:** 2022-11-28

**Authors:** Crystal Kwan, Ho Chung Tam

**Affiliations:** The Department of Applied Social Sciences, Hong Kong Polytechnic University, Hong Kong, China

**Keywords:** older people, living alone, in-poverty, Hong Kong, qualitative study

## Abstract

Despite the growing number of older people who live alone and in poverty, the intersection of these two social risk factors and the impacts on older adults’ health and well-being have not been widely examined. This qualitative study explores the challenges and strengths of 47 older people who live alone and in poverty in Hong Kong. Thematic analysis was used to identify eight themes related to challenges: (i) social isolation and loneliness, (ii) self-esteem and self-efficacy, (iii) declining mobility, health and activity levels, (iv) high medical expenses, (v) age discrimination and long wait times for medical health services, (vi) age discrimination, retirement, and wanting part-time employment, (vii) not enough gender-specific social participation activities, and (viii) housing insecurity. Four themes related to strengths were identified: (i) An “I have enough” mindset, (ii) strong formal social support, (iii) contributing to the community and others, and (v) “Most of us like to be alone.” Successfully addressing poverty in old age and tackling the challenges associated with living alone will require focusing on and activating not only external and systemic resources but also the intrinsic capacities and strengths of older adults themselves. Five discussion points are raised addressing the implications for future gerontological research and practice.

## 1. Background

### 1.1. Older People Living Alone

The number of older people living alone is rising, in Hong Kong [1], and elsewhere around the world, such as Canada and the United States [2], the United Kingdom [3], Japan [4], and South Korea [5]. Changing family structures and increases in life expectancies are contributing to this global phenomenon [6]. The World Health Organization [7] identifies living alone as a social and health risk factor for older adults. Research indicates that older adults who live alone are more likely to have worse health outcomes than are their counterparts who live with others. For instance, older people who live alone are more likely to have high risk of disabilities [8], chronic diseases [9], mental health problems, [10], malnutrition and unhealthy dietary habits [11], challenges to activities of daily living (Instrumental/Activities of Daily Living), [12] and mortality [13]. Furthermore, this group of older adults is likely to experience higher levels of financial strain, social isolation, loneliness, smaller social networks, less emotional support, and lower quality of life than their peers who live with others [14]. Living alone and feelings of loneliness have been intertwined in the literature [15]. Although they are inter-related, studies have found that living alone is independently associated with higher risk of mortality [16], and have differential effects on health. For example, Beller and Wagner [15] found in their longitudinal sample of the German Aging Survey (N = 4184), that living alone was a better predicter of physical and cognitive health whilst subjective loneliness was a better predictor of mental health.

### 1.2. Old Age Poverty

In tandem with the growing number of older people living alone is the number of older people living in poverty, which is also rising, not only in Hong Kong but globally. Despite enhanced overall living standards and wealth in many developed economies, old age poverty persists. For instance, the latest rates of old age poverty in Japan, the United States, Australia, and Korea, were 19.6%, 19.6%, 23.2%, and 43.8%, respectively, while those countries’ GDPs per capita during the same time were relatively high [17]. Poverty is a social determinant of health that can limit access to resources, such as safe/stable housing, education, employment, and food and nutrition that support good health and well-being [18].

Hong Kong has a persistent and growing problem of old age poverty. From 2010 to 2017, Hong Kong experienced, on average, an annual GDP per capita growth rate of 2.09% [19]. In contrast, the proportion of older adults living in poverty in Hong Kong has significantly increased over recent decades [14,20], from 27.7% in 1991 [21] to 44.4% in 2017 [20]. Regardless of what measure of poverty is used (e.g., income, social exclusion index, material deprivation index, and asset poverty measures), studies have found that the poverty rate is higher among older adults in Hong Kong than it is in any other demographic [14,21,22,23].

Old age poverty has adverse consequences from the micro level to the macro level. Specifically, it has been associated with various negative physical and mental health outcomes for individuals [24,25] and results in greater public expenditures on healthcare and social services, larger constraints on GDP and economic growth, and widening of the societal inequality gap [26]. As societies worldwide experience unprecedented rates of population ageing (including Hong Kong, whose older population is expected to represent 31.1% of the total population, or 2.37 million people, by 2036 [27] and the risk of falling into, staying in, or sinking deeper into poverty may increase with old age (a result, for example, of inadequate pension benefits, limited employment opportunities, and age-related chronic health conditions), efforts to address this poverty paradox require timely research and attention focused on improving practices and policies.

### 1.3. Older People Living Alone and in Poverty: The Intersection

Living alone and in poverty are associated with negative health and well-being outcomes for older adults [8,9,10,11,12,13,14,24,25]. In Hong Kong, studies have indicated that older adults living alone are not only more likely to be income-poor, but also asset-poor, socially excluded, and materially deprived [14,22,23,28]. Additionally, the decrease in the number of wage earners in a household increase the risk of poverty. Despite the growing number of older people living alone and in poverty [1,2,3,4,5,14,17,20], the intersection of these two social risk factors and the impacts on older adults’ health and well-being have not been widely examined, and the limited existing research has been oriented narrowly toward the challenges and negative outcomes associated with living alone and in poverty.

Although older adults who live alone and in poverty certainly face various systemic disadvantages, they also have intrinsic strengths that have been largely unexamined but are important to identify and to utilize in the development of practices and policy interventions that support their health and well-being. For instance, Kok et al.’s [29] qualitative study examining the successful ageing of older adults in a low socioeconomic position, found that the participants “possessed multiple individual and social resources that enabled them to deal with various adversities throughout their lives” (p. 851). Individual resources were related to the older adults’ personality, values and attitudes, and included for example “valuing one’s own skills”; “valuing other aspects of life more highly than social status”; “emphasizing gradual improvements in living conditions”; “preserving” and “resigning oneself to adversity” (p. 847). Importantly, their study findings counter the binary stereotypes of older adults living in poverty as helpless victims or “heroic images of resilience” (p. 843), and shed light into a more nuanced understanding of older adults living in poverty.

### 1.4. Research Aim

The paper addresses the knowledge gap and presents the findings from a research project that explored the dynamics of poverty across the lifespan of elder-headed households, including older people living alone and those living only with a spouse. The study’s findings relating to the participants and data relevant to the older people living alone are reported here. Data were extracted and analyzed to explore the challenges and strengths of this subgroup of older adults. The findings identify the target population’s challenges and strengths and propose ways that those factors can inform interventions to enhance the quality of life for older adults living alone and in poverty.

## 2. Methods

This study was approved by the Human Subjects Ethics Review Board at the Hong Kong Polytechnic University (Reference Number: HSEARS20200113002). In addition to procedural ethical protocols, we also developed a distress protocol adapted from Draucker et al. [30] to follow in the case where participants through the interviews may feel considerable negative reaction/distress, which included strategies such as how to build rapport with the participants, use of person-centered calming techniques to diffuse situations, consistent monitoring of participants’ emotional reactions, debriefing post-interview check-in and providing information on available psychological or social services.

We adopted a qualitative approach that used life history interviews [31] of 47 older adults living alone, with interviews lasting 1 to 1.5 h. The participants all spoke Cantonese or Mandarin, and the interviews were conducted in their preferred language. The interviewer (second author) is fluent in Cantonese, Mandarin, and English. The interviews were transcribed verbatim into English. For translation transparency and rigor [32], the transcriber retained the Chinese characters for words or phrases that did not have a direct translation into English and in brackets explained the context or meaning of the word or phrase in English.

Data relevant to the participants’ late life experiences were extracted and analyzed for this paper. The interview topics are included in Appendix B. This study employed the six-stage method of thematic analysis (TA), with a hybrid approach to coding whereby possible codes were derived from the literature [33,34] and then line-by-line coding and the constant comparison method were used [34]. An initial codebook was developed with each of the analyst independently analyzing five of the same transcripts and then coming together to discuss and resolve any discrepancies. The codebook would be updated once another five transcripts were analyzed. Both authors were involved in the analysis, the details of which (including theme identification) are given in the Appendix A.

### Participant Recruitment and Characteristics

A local NGO service provider for low-income citizens helped us to recruit the participants, who had to meet the following inclusion criteria: (i) age 65 years or older, (ii) living alone, (iii) being low-income, on the basis of the Hong Kong official poverty threshold, (iv) residing in Hong Kong, and (v) not residing in owner-occupied housing (and not owning a home). Table 1 lists the details of the participants’ characteristics.

## 3. Findings

### 3.1. Common Challenges

Six key themes were common across the challenges reported by the majority of the participants.

#### 3.1.1. Key Theme 1: Social Isolation and Loneliness: “What If I Die and No One Notices?”

One of the most frequently discussed challenges was social isolation and loneliness. “What if I die and no one notices?” is an actual quote from a participant (Female, 74). Another participant explained:


*I live alone. Who will know if something bad happened to me? If nothing happens, that’s great. But if SOMETHING happens… it would be pathetic. Sometimes I go home from the hospital, and I feel extremely tired, but I still need to cook for myself. We older people who live alone worry about many things.*
(Female, 74)

Another echoed this sentiment, saying:


*What I am saying is I have diabetes, and if I fainted no one would know, and I will probably die. I am all alone. At times I think to myself: I will die, and no one will know. Everyone has their life, and no one would bother to care about me. I’m happy, but two things I fear are the house [referring to the possibility of losing public housing] and dying alone.*
(Male, 69)

Participants also shared how their social isolation and loneliness were expressed in their everyday life routines: *“Living alone…is troublesome, no one talks with you. After having meals, I sleep, I watch TV, and then sleep again after watching TV, just like this, nothing, that’s it—you know?”* (Female, 88) Another participant echoed this sentiment, saying:


*Living alone, you can’t even find someone to chat with. If something happens, even mundane things like my scratching an itch on my back, no one can help you with that and you can’t help yourself, either. Earlier I got the shingles, I ha[d] to beg a few times before someone would help me to apply medicine to my back.*
(Male, 65)

#### 3.1.2. Key Theme 2: Self-Esteem and Self-Efficacy

When asked about their perspectives and ideas about support for their well-being, the participants often responded that they had little to contribute because they lacked formal education. Common responses included: *“I don’t know stuff like that.”* (Female, 78) Another participant shared the reason why he lived in subdivided flat (a substandard housing situation, which is a part of a flat that is further subdivided, with the living space approximately just 40 square feet, and with poor ventilation and hygiene [35]): *“Firstly, I’m poor. Secondly, I’m not very educated. So, I get to live in places like this, right? I don’t know a lot of words.”* (Male, 65) *Another reported: “I’m nervous and bad at talking, and uncultured and illiterate, you know? That’s what I meant when I said I’m uncultured. I don’t know stuff.”* (Female, 72) When asked to identify what support is needed, a participant replied: *“We need to feel good about ourselves.”* (Female, 75).

#### 3.1.3. Key Themes 4: Declining Mobility, Health, and Activity Levels

Declining mobility, health, and activity levels were other key challenges identified by the participants. As one participant shared:


*In recent years, my knees started to hurt a lot and my relative suggested [I] go to a nursing home. She told me to apply when the situation is not that bad because it takes a few years of being on the waiting list. It’s hard to tell when it [her knees] would deteriorate. I am worried about that, so I applied.*
(Female, 80)

Other participants worried that such declines in mobility and health would cause them to go out into the community less. One participant described:


*I seldom go out. I want to, but I can’t. Many things hinder me. I can’t walk fast. My knees make me have trouble in walking up stairs. I am scared of the stairs, so I don’t dare do many things. If you can still walk, you can find someone who can help you. If you want to talk to somebody, you can go out and find someone. How could this be achieved if you can’t walk?*
(Male, 71)

Another participant echoed that sentiment:


*I joined as a volunteer with the church to do some home visits with other older adults, but it is difficult for me to walk, I feel exhausted when I go upstairs and downstairs. I want to prevent further damage on my joints, as there is wear and tear on them. There is no other way but to have a joint replacement. I am not willing to receive the replacement, so I stopped joining those activities.*
(Female, 77)

#### 3.1.4. Key Themes 4: High Medical Expenses

Another key challenge the participants identified involved concerns over current and future medical expenses––a particular worry for older adults who had multiple health issues (e.g., “the three highs”––high blood pressure, high blood sugar, and high cholesterol/fat) and were on multiple medications. Despite access to healthcare vouchers (a government subsidy of HK$2000 for all older adults [65+] residents [36], the older adults shared how their medical expenses remained high because the vouchers either did not cover enough or did not cover a specific need. As one participant shared:


*Normally manageable [referring to his living expenses], but if I am sick or need to go to the hospital I don’t have spare money for that. It’s a minimum of 500 dollars each time. I need to keep watching my budget for that.*
(Male, 89)

Another participant shared:


*I have the Health Care Vouchers, but I have used all of them. Now I pay for it myself. It costs around 300 dollars each time I visit the doctor. It is expensive. Sometimes I just stick a band-aid on instead of seeing a doctor. It can’t eliminate the pain completely, but I will feel more comfortable.*
(Female, 70)

Interestingly, despite their low income, some participants reported a willingness to pay extra for private care (including Chinese traditional health methods that are not covered by the vouchers). One participant shared: *“The dit da (a Chinese medicine to treat bone-related injuries) for bad knees costs tens of dollars each. What else is left at the end of the month?”* (Female, 74)

#### 3.1.5. Key Themes 5: Age Discrimination and Long Wait Times for Medical Health Services

Experiencing age discrimination and/or long wait times for medical health services was another key theme in the challenges experienced by the participants. One participant shared:


*He [the doctor] said it was because of degeneration. This is what the public hospital doctors do: old people come to them for knee pain, and they say it is untreatable degeneration.… they did not even assess me. They said the degeneration is irreversible!*
(Female, 80)

Another reported:


*But for the x-ray, I need to wait for 5 years because I am 75 years old now. I am too old. I still did not have the scan even now. This is a long-term knee pain, I don’t have the knowledge of what to do, so I just tolerate the pain.*
(Female, 75)

A third participant shared:


*I am most concerned about my nose. I cannot sleep when I have a stuffy nose, but no doctor helps me …. I have waited for 3 years. Is it reasonable?*
(Female, 74)

#### 3.1.6. Key Theme 6: Age Discrimination, Retirement, and Wanting Part-Time Employment

Participants also reported experiences of age discrimination regarding employment. For instance, many shared that, despite wanting to continue to work, they felt they were pushed to retire due to their age. One participant explained:


*It was not “retirement.” It was because I can’t walk fast. My knees are sore. I can’t work efficiently like others—what could I do when I only closed three deals as others closed five deals? [I was] sixty-something. I can’t work efficiently. Gradually, I had conflicts with my colleagues. They felt it was unfair when everyone took the same salary while I did less. The head [of staff] also complained to me. So, I was half-retired at that time.*
(Male, 71)

Another shared:


*I was the chief writer of [name of newspaper]. The boss was rubbish. We could not get along. When I was first recruited, he promised that I would have my share. After the newspaper became well-known, he bought himself an industrial building. Once I reached 65 years old, he kicked me out and hired people who just graduated from university so he could pay less. Sneaky rubbish.*
(Male, 90)

Furthermore, the participants stated that they still wanted to work, especially part-time, but struggled to find employment due to their age. One shared: “*Yeah. I thought about going back to work after he [her husband] died—washing dishes or something else. But they don’t hire people over sixty.”* (Female, 91) When asked if she currently works, another participant shared, *“No, [laughs] who would hire me?”* (Female, 75).

### 3.2. Challenges Unique to Some Participants

There also were two unique challenges experienced by some of the participants.

#### 3.2.1. Unique Theme 1: Not Enough Gender-Specific Social Participation Activities

Among the small number of older men interviewed, some shared that they did not participate in specific activities because they felt the activities were strongly geared toward older women. One participant explained why he switched locations to do his exercise: *“Originally it was in [neighborhood A], and there are a lot of women crowded in the room, so I go to the [neighborhood B] one instead.”* (Male, 83) Another participant opined, *“Majority of females love gathering and chatting. Men don’t. Men are lonelier and more egocentric, so they don’t get on well together.”* (Male, 73)

#### 3.2.2. Unique Theme 2: Housing Insecurity

The majority of the low-income participants were living in public housing estates, with a small minority either on a waiting list or not yet qualified for public housing. For those participants, housing insecurity (along with health worries) was the primary concern. As one participant shared:


*Sometimes I can’t sleep, thinking about this [referring to his housing insecurity]. Yeah, sometimes I drink one can of beer to sleep. I used to not drink. But I have some beers in my fridge, and I’ll open one when I can’t sleep. I will not take sleeping pills.*
(Male, 69)

Another participant explained his housing situation:


*Yeah, a lot better [referring to public housing estates]. Subdivided units are a lot different from public estates. It’s much narrower and constraining. You enter a flat and it’s divided up to ten units. My life right now, to be frank, living in a subdivided unit, there’s a lot of bed bugs. They are in all the rooms. We have to spray insecticide all the time. Also, they usually come out at night. They wait for you to fall asleep for your blood. So, I write calligraphy till late, often until dawn before I sleep, and when I see a bed bug I catch it. I catch them all before I go to bed.*
(Male, 78)

Another participant shared that despite a bedbug problem, she was unable to move because her current building had a lift, which she needed due to her limited mobility:


*… more bugs in these two years, especially this year! I cannot sleep well. People keep moving in and out, but I cannot. I don’t know where I can go. There is a lift in this building, I cannot walk up and down the stairs due to my knees. My friend introduced this flat to me because there is a lift, I don’t need to walk stairs.*
(Female, 74)

### 3.3. Common Strengths

Three key themes of strengths were common across the majority of the participants.

#### 3.3.1. Key Theme 1: An “I Have Enough” Mindset

When asked about future expectations as they aged, a common theme was the mindset of having enough. As one participant shared:


*I never ask too much. Living a mundane life: healthy, enough food, enough money. I don’t spend on luxurious things. So far, that’s it. Now I wait for nothing. I need nothing. The future? Not much. ‘Come quietly and leave peacefully’, I guess. [laughs] I am eighty and I am on the way to the “end of road.”*
(Male, 83)

Another, who had experienced the Cultural Revolution and war, shared:

*The condition back then was not good, and my grandpa was part of the “Five Black Categories” [during the Cultural Revolution, there were five groups that were persecuted: landlords, rich farmers, counter-revolutionaries, bad influencers, and right-wingers* [37]]. *We have suffered many hardships, so I cherish my current life. Happiness consists in contentment; it is very important for everyone. Even having weak tea and scanty meals, I am satisfied with them.*(Female, 73)

Yet another, who also had experienced the same historical contexts, pointed out:


*Everyone lived very poor in the past. We didn’t get to go to yum cha all the time like now. You had to be very wealthy with a house to enjoy the luxury of going to yum cha. We didn’t have three meals a day. Two meals a day is already lucky. One meal at 9 am, and another at 4 pm. Who gets three meals a day? I am over 80, I’ve seen everything. The Japanese invading, and Cultural Revolution. Now, now is the most peaceful time.*
(Female, 86)

One participant even shared the following in response to the recent pandemic consumption vouchers (a government subsidy to all residents during the pandemic): *“I would wish that they don’t give us so much! They are giving us HK$2000 again, soon right? The more money you spend the more you want to spend.”* (Female, 75).

#### 3.3.2. Key Theme 2: Strong Formal Social Support

Participants also reported that their formal social support network was strong, noting that there were no shortages of NGOs, public community workers (e.g., councilors), and community organizations reaching out to them, offering social participation activities and support to meet their basic needs (e.g., packages of rice, free lunch meals). One participant shared: *“The NGOs are really kind. They give us free meals every Monday, Wednesday, and Friday. On Tuesday, they also give us food, like bread.”* (Female, 73) Another shared that her local district councilor had helped her find a job and helped with other applications after her husband died:


*The district councillor––He was quite nice. He recommended me to sweep the floor [referring to her cleaning job]. When I encounter difficulties, I can seek advice from him. I don’t know words well and he helped me to fill the form to apply for the Consumption Voucher Scheme [a government subsidy during the pandemic]. Their office is on the 3rd floor below us. Everyone treats me well.*
(Female, 70)

Another participant described receiving help with material goods during the different seasons: *“The building next to us is called [Name] Building. [Name] Welfare Association is there, and they care about me a lot. Last year, they gave me a blanket. It was very cold that year. This year it is hot, and they gave me a fan cooler.”* (Female, 72)

#### 3.3.3. Key Theme 3: Contributing to the Community and Others

Participants also reported contributing to the community and others in various ways. One participant described accompanying her neighbor to the hospital:


*I offer help whenever it is within my power to. Like the one downstairs. She asks me to go to the hospital with her, so I accompany her. Once she had a surgery, I bought food for her for three months. I help out.*
(Female, 74)

Another participant shared a similar situation:


*There is one who just got into [name] Hospital. He is 84 years old, living in the [name] building. He has chronic illnesses. Last year, I was walking out[side] the building and saw him having shortness of breath, so I took him to the hospital. After he was hospitalized, I got a call saying he had a stroke. He was in a coma for over ten days, and he woke up. He had no one to turn to, so he could only call me. I do what I can do. It took about two months for him to recover. The period before he was discharged, he cried a lot to me, saying he doesn’t know what to do, whether he should go to a nursing home or not. I had no idea of what to do, so I sought help from a doctor from the place [where] I do volunteer work, in [Name] Hospital. Then [Name of Social Worker] recommended him to a nursing home. He was lucky having me to help him—of course [name of social worker] also helped a lot.*
(Female, 74)

Another participant reported that she volunteered to deliver meals to homebound older adults: *“I usually do volunteer work here. I deliver meals on Sundays.” (Female, 87)* Another participant shared how she helped out during the pandemic: *“Now I contribute to the society. I join voluntary work when I have time. During these two years, we packed face masks and materials, and delivered them to other older people.”* (Female, 73)

### 3.4. Strengths Unique to Some Participants

#### Unique Theme 1: “Most of Us Like to Be Alone”

A small number of participants shared preferences for living alone. As one participant described:


*I don’t feel so sad. Right now, I feel good being alone. I have way less burden. Except if you really have someone you really love who you want to spend your life with. Or else I really don’t want to become others’ burden. There’s nothing wrong about being alone, we all come here on our own, you brought nothing, so you don’t need to bring anything away with you either when you leave, right?*
(Male, 78)

Another participant shared:


*I think it’s more freedom for myself. No one to make me angry, nothing to see, right? If I have friends over we can drink tea if we want to drink tea, watch a movie if we want. Walk around if we want. I think it’s freedom.*
(Female, 74)

A third participant offered this perspective: “*Most of us like to be alone. [laughs]”* (Female, 75).

## 4. Discussion

The study’s findings present important insights for future research and practices in Hong Kong, and can be extended to other contexts that also are witnessing a greater proportion of older adults living alone and in poverty. Thus, providing policy and practice insights locally and globally on social and cultural determinants of health. Five discussion points arose from the insights.

First, the findings illustrated that medical health challenges (especially mobility problems) are a major concern for this group of older adults, and at the same time they are motivated to improve their health, with many being willing to pay for private healthcare (including alternative traditional Chinese methods not covered by the health care vouchers) when the public system dismissed their needs or failed to address them, despite their low income and frugal lifestyle. Indeed, the high medical expenses related to such care (or future care) are a barrier to escaping poverty and can even deepen it. Addressing old age poverty must include a deeper examination of the role of public medical health systems. Future research should explore how age-friendly interventions within public medical health systems can impact poverty in old age.

Second, the findings suggest that a subgroup exists within this already marginalized group and is at a greater disadvantage, thus requiring additional support: individuals who are not living in public housing estates due to ineligibility or being on a waiting list. This subgroup is repeatedly falling through the systemic gaps. In Hong Kong, as of March 2022, the average wait time for an older adult one-person applicant was 4.1 years [38]. During that time, the individuals live in substandard circumstances (e.g., subdivided flats) that can further exacerbate health issues (e.g., depleted sleep from bed bugs at night). The costs of such flats can be 80% to 90% of their already low incomes (e.g., a subdivided flat for one of the participants costs HK$ 4000/month and CSSA is HK$4250/month), leaving almost nothing for day-to-day expenses, such as food. The housing crisis in Hong Kong is a complex and long-term challenge, but in the short and middle terms, older adults who live in poverty and alone, and in precarious housing situations, should be prioritized to receive additional support, such as social support, material needs, and activities, as they are at a high risk of homelessness and worsening of health. Strong empirical evidence indicates the negative impacts of poor housing and financial stress on health [39]. Future research should explore the deep intertwining of old age poverty and homelessness, because both are growing problems in Hong Kong [40] and in other contexts as well, including, for example the United States [41], Canada [42], and England [43].

Third, the findings shine a spotlight on employment among older adults. Current discourse regarding the older adult workforce is expansive, as societies are experiencing rapidly ageing populations [44]. Nevertheless, retraining programs often exclude older adults who have little to no formal education and have worked in “low-skilled” and precarious jobs their whole lives. Furthermore, in an increasingly digitized economy, the digital divide is ever more present for this group, whereas the findings illustrate that these older adults want employment. Thus, innovation is needed to examine what meaningful employment for these older adults can look like. A focus on exploring their strengths would be a first step––especially their “soft” skills and life experiences. For example, many of the participants shared about contributing significantly to their community, especially by supporting their neighbors who are also older adults but require extra care and support. Such work could be recognized as community health work. Programs that use the strengths of older adults and train them (especially those who are healthier and more able) to be part-time community health workers within their own communities could be explored, to help address the challenges experienced by this group. Further, in some contexts, like Canada, such community-based lifelong learning programs for older adults at risk are publicly subsidized and have found to be an “important social determinant of health in an aging society” (p. 689) [45].

Fourth, the findings suggest gender differences to be explored further. First, there is a need to more deeply examine the barriers and facilitators to social participation for older men who live alone and in poverty. The findings relating to gender differences in this study come from a very small sample of older men and cannot be generalized. However, those findings do align with other research. For instance, [46] found, in their longitudinal study using a national dataset of older adults living in Taiwan, that before retirement men displayed higher social participation than women, “but as age advanced, social participation decreases for men while women experience an increase” (p. 61). Future research examining gender differences in social participation should include subgroups of older men, especially those who live in alone and in poverty and therefore may face additional barriers (e.g., financial strain) to participation. Second, as the majority of the participants are female, this may suggest that the key challenges identified in the study (e.g., age discrimination, self-esteem, self-efficacy, social isolation and loneliness in late life) have a gendered dimension. There is empirical evidence that indicate older adult males are more likely to have higher levels of self-esteem (which is a predictor of life satisfaction) than their female counterparts [47]. Future research could explore how gender (specifically being female), age, and cultural context intersect to implicate discrimination, self-esteem, self-efficacy, social isolation and loneliness in late life. Finally, the findings related to the “I have enough” mindset offer an opportunity to deepen intergenerational relationships. In today’s sociocultural context, materialism and the thirst for hyper productivity and progress can contribute to a toxic attitude and mindset of “never having enough” [48], because enough is an ever-elusive moving target. Younger generations, who lack lived experiences of world wars and extreme poverty, may be more vulnerable to such mindsets and their negative impact on mental health. A counter movement, called minimalism, has captured the attention of millennials and Gen-Xers as a valuable lifestyle to embrace [49,50]. The older adults of this study are model minimalists and can offer much through sharing their mindset and lifestyle. Their orientation toward minimalism can help bridge the need for stronger intergenerational communication and relationships, and can teach younger generations a great deal. This finding (“I have enough”) also act as a point of reflection for us as researchers, who adopt a healthy ageing framework [51], which has emerged from largely Western perspectives that prioritize functionality and productivity. This theme is a reminder for us as researchers to step back from our dominant framework/lens and to recognize alternative perspectives of what constitutes health and wellbeing in late life. Another theme related to this mindset is the “most of us like to live alone” theme, which indicates living arrangement preferences that are worthy of further exploration. Culturally, living with multiple generations is an ideal preference for Chinese older adults, however there is research to indicate a shift. For example, in Meng et al.’s [52] nationally representative sample of older people in Urban China, examining future living arrangement preferences of middle-aged and older people that living close to their children (e.g., in the same neighborhood) was the most popular preference, with living with adult children coming in second.

## 5. Conclusions and Study Limitations

The number of older adults who live alone and in poverty is increasing, in Hong Kong and worldwide. An understanding of their perspectives and lived experiences is necessary in order to better support them. Such personal experiences of ageing are undervalued in empirical research but are important in transforming older adults from mere objects of study to active participants in research. Although this qualitative study cannot be statistically generalized to all older adults living alone and in poverty, it provides in-depth and nuanced insights that elucidate the gaps in knowledge, practices, and policies. Our findings stressed that older adults who live alone and in poverty face numerous challenges, but we were limited in identifying the causes of such challenges. Still, the study highlights the interconnections among the challenges. For instance, age discrimination and long wait times for medical health services may erode older adults’ mobility, health, and activity levels, and that in turn could exacerbate their social isolation, loneliness, and poor mental health, self-esteem, and self-efficacy. The findings about the challenges reaffirm previous research, and the results regarding older adults’ strengths warn against viewing this group through simply a deficit lens and as passive recipients of care. Addressing poverty in old age and tackling the challenges associated with living alone will require focusing on and activating not only external and systemic resources but also the intrinsic capacities and strengths of older adults themselves. Lastly, this study illustrated successfully addressing health inequalities among older people, will necessitate a focus on social (e.g., living alone and in poverty) and cultural (e.g., sociocultural values and generational differences) determinants of health.

## Figures and Tables

**Table 1 ijerph-19-15856-t001:** Participant Characteristics.

Characteristic	Participants (*n* = 47)
**Gender**	
Female	37
Male	10
**Age**	
Young-old individuals (age 65–74 years)	5
Middle-old individuals (age 75–84 years)	23
Old-old individuals (age 85+ years)	19
**Place of birth**	
Hong Kong	10
China (Mainland)	32
Other ^1^	5
**Length of time living in Hong Kong**	
Less than 10 years	2
11–20 years	2
21–30 years	6
31–40 years	7
41–50 years	5
50+ years	25
**Language spoken**	
Cantonese	44
Mandarin	3
**Marital status**	
Divorced	9
Never Married	3
Married ^2^	4
Widowed	31
**Housing type**	
Private flat (rental)	1
Subdivided flat	4
Subsidized public housing	37
Village house	5
**Age of receiving public housing**	
Adulthood (up to age 44 years)	6
Late middle age (age 45–64 years)	16
Older adulthood (age 70–79 years)	14
Not applicable	11
**Family members**	
No children	9
1 child	9
2 children	13
3 children	8
4 children	3
5 or more children	5

^1^ Other = Indonesia, Macau, The Philippines, and Vietnam. ^2^ Married = the participants were living separately from their spouses because of separation, or because the spouses were living in Mainland China or a nursing home.

## Data Availability

Data supporting reported results can be given upon request of the first author. Analytic methods are described in the Appendix A.

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
