# Peer review of "“What If I Die and No One Notices?” A Qualitative Study Exploring How Living Alone and in Poverty Impacts the Health and Well-Being of Older People in Hong Kong"

_ijerph, 2022, doi:10.3390/ijerph192315856_

Round 1

Reviewer 1 Report

Title:

“What if I die and no one notices?” A Qualitative Study Exploring how Living Alone and in Poverty impacts the Health and Well-being of Older people in Hong Kong 4

Comments: There is an error in the title in the word Qualitative, a ta is missing.

The aim of the present study was to explores the challenges and strengths of older adults living alone and in poverty in Hong Kong. The manuscript has several strengths. However, it has major shortcomings limit its potential.

1.     Abstract: It is clear, however it does not include the data analysis approach. Moreover, one of the name of the themes (men-specific), seems quite sexist and authors should be aware of the usage of a inclusive language. 

2.     Introduction: The introduction needs more evidence, the subheading “Older People Living Alone and in Poverty” lacks of evidence or citations that support the statements, and it will be useful to include the current evidence that contradicts that living alone it is different from loneliness.

3.     Methods:

3 .1 Design and procedure. The study used a life history interview approach which seems appropriate. 

3.2.Participants. Even though the participants section is clear, I recommend to include  subheading as it is confusing.

3.3.Interview: The authors did not describe the interviews sections what themes they explored.    

3.4. Data analysis: The researchers did a six-step thematic analysis, which seems appropriate. Even though in the supplementary material the analysis is well explained, the manuscript will be improved if the authors give more detail about the analysis in the methods section. The authors do not discuss ethics.

4.     Results. The results are interesting and important, nonetheless are extremely long, therefore it gives the sense that some themes might belong together or more abstraction in the analysis is needed.

5.     Discussion. The discussion is very well written and interesting.  

My major concerns are:

1.     Methods: more details and information should be provided.

2.     The results are extremely long, therefore it gives the sense that some themes might belong together or more abstraction in the analysis is needed.

Author Response

Dear Reviewers,

Thank you for taking the time to review and consider our manuscript. We appreciate the detailed and thoughtful feedback. Below in blue text we outline the changes made to address each of the reviewers’ comments.  

Reviewer 1

Comments: There is an error in the title in the word Qualitative, a ta is missing.

Response:

The title has been corrected and reads: “What if I die and no one notices?” A Qualitative Study Exploring How Living Alone and in Poverty impacts the Health and Well-being of Older people in Hong Kong”

The aim of the present study was to explores the challenges and strengths of older adults living alone and in poverty in Hong Kong. The manuscript has several strengths. However, it has major shortcomings limit its potential.

  1. Abstract: It is clear, however it does not include the data analysis approach.

Response:

The sentence on lines 13-14 was revised to identify thematic analysis as the approach to identify themes: “Thematic analysis was used to identify nine themes related to challenges: …”

Moreover, one of the name of the themes (men-specific), seems quite sexist and authors should be aware of the usage of a inclusive language. 

Response:

Changed the term “men-specific” to “gender-specific” in the abstract and throughout the manuscript.

  1. Introduction: The introduction needs more evidence, the subheading “Older People Living Alone and in Poverty” lacks of evidence or citations that support the statements, and it will be useful to include the current evidence that contradicts that living alone it is different from loneliness.

Response:

In subheading 1.2 “Older People Living Alone and in Poverty: The Intersection” we have added the relevant citations to support the statements within that paragraph:

“Living alone and in poverty are associated with negative health and well-being outcomes for older adults [8-14, 24, 25]…. Despite the growing number of older people living alone and in poverty [1-5, 14, 17,20],…”

We added the following text in this subheading as well:

For instance, Kok et al.’s [32] qualitative study examining the successful ageing of older adults in a low socioeconomic position, found that the participants “possessed multiple individual and social resources that enabled them to deal with various adversities throughout their lives” (p. 851). Individual resources were related to the older adults’ personality, values and attitudes, and included for example “valuing one’s own skills”; “valuing other aspects of life more highly than social status”; “emphasizing gradual improvements in living conditions”; “preserving” and “resigning oneself to adversity” (p. 847). Importantly, their study findings counter the binary stereotypes of older adults living in poverty as helpless victims or “heroic images of resilience” (p. 843), and shed light into a more nuanced understanding of older adults living in poverty.

The following was added to the reference list:

  1. Kok, A. A. L.; van Nes, F.; Deeg, D. J. H.; Widdershoven, G.; Huisman, M. “Tough Times Have Become Good Times”: Resilience in Older Adults With a Low Socioeconomic Position. The Gerontologist. 2018, 58(5), 843–852.

  1. Methods:

3 .1 Design and procedure. The study used a life history interview approach which seems appropriate. 

3.2.Participants. Even though the participants section is clear, I recommend to include  subheading as it is confusing.

Response:

Added a subheading on p.3 “2.1 Participants Recruitment and Characteristics”

3.3.Interview: The authors did not describe the interviews sections what themes they explored.    

Response:

The interview topics explored are in the supplementary materials, so we added the following sentence to be clearer on p. 3 “The interview topics are included in the supplementary materials.”

3.4. Data analysis: The researchers did a six-step thematic analysis, which seems appropriate. Even though in the supplementary material the analysis is well explained, the manuscript will be improved if the authors give more detail about the analysis in the methods section. The authors do not discuss ethics.

Response:

The following text was added to the main body of text (p. 3) to provide more details of the analysis: “An initial codebook was developed with each of the analyst independently analyzing five of the same transcripts and then coming together to discuss and resolve any discrepancies. The codebook would be updated once another five transcripts were analyzed.” Further, there is the following sentence that tells the reader greater details of the analysis procedure, including theme identification are given in the supplementary materials: “…the details of which (including theme identification) are given in the supplementary materials.”

The following text was added on p. 3 in the methods section to further elaborate on the ethical protocols of the study: “In addition to procedural ethical protocols, we also developed a distress protocol adapted from Draucker et al. [33] to follow in the case where participants through the interviews may feel considerable negative reaction/distress, which included strategies such as how to build rapport with the participants, use of person-centered calming techniques to diffuse situations, consistent monitoring of participants’ emotional reactions, debriefing post-interview check-in and providing information on available psychological or social services.”

The following was added to the reference list:

  1. Draucker, C.B.; Martsolf, D.S.; Poole, C. Developing distress protocols for research on sensitive topics. Archives of Psychiatric Nursing. 2009, 23(5), 343-350.

  1. Results. The results are interesting and important, nonetheless are extremely long, therefore it gives the sense that some themes might belong together or more abstraction in the analysis is needed.

Response:

We have removed the theme “mental health” – as it is a bit general (as compared to the mental health related specific themes of “social isolation and loneliness” and “self-esteem and self-efficacy”) and the contents coded in this theme can be collapsed into more existing specific themes. We also deleted three quotes under the “I have enough” mindset theme (under common strengths), as this theme had too many illustrative quotes (7 compared to the other themes). Aside from those revisions, we kept the remaining results section intact, as the qualitative approach is aimed to capture a nuanced and in-depth/rich understanding of the participants, and the specificity of the themes and illustrative quotes accurately depicts the complexity and nuanced understanding of the phenomenon.

  1. Discussion. The discussion is very well written and interesting.  

My major concerns are:

  1. Methods: more details and information should be provided.
  2. The results are extremely long, therefore it gives the sense that some themes might belong together or more abstraction in the analysis is needed.

Response:

Please see the responses above as the address these two major concerns.

Reviewer 2 Report

The authors set to discuss and analyse very timely and important issues related to poverty, loneliness and ageing. This qualitative study explores the challenges and strengths of poor older adults living in Hong Kong. The authors focus on a variety of aspects that include social isolation, mental health, self-esteem, activity levels, medical expenses, ageism, retirement, and housing insecurity, among others. They identify the strengths and discussion points for future gerontological research and practice, and give solutions on how to improve the quality of life for older adults living alone and in poverty. The authors provide an updated and relevant lit review on the issues presented in the paper and outline the most important risk factors associated with ageing in poverty and being alone, by taking into account both the micro and macro levels. The paper is well written and clearly presented. I have some recommendations, though, that could improve it: 

It would be interesting to explore in more detail the gender dimension and discuss whether women, who are aged by culture earlier than men and subject to the double standard of ageing, present different perceptions on ageing, self-esteem, and ageism, and social isolation. This is important as older women live longer and are more prone to old age loneliness and ageism, and, thus, may need different actions to improve their wellbeing and quality of life. More specfic solutions for older women could be consided, if possible, by taking into account the particular cultural and social context (Hong Kong). 

Related to the findings on employment among older adults and suggested innovative solutions, the authors might consider life-long learning programmes and intergenerational relations. They already mention them, but could elaborate a bit more: see similar articles on intergenerational benefits in relation to old age.

To improve wellbeing of older people, the authors could also reflect on more personal experiences of ageing, which are often ignored and undervalued in empirical research. They could mention storytelling or narrarives on personal ageing experiences, which make older adults not the mere objects of study, but rather active subjects in research on health care and social policy services. 

Also, it would be interesting to see how sociocultural factors and contexts, in this case, living in Hong Kong, alter the perceptions of ageing, living, and dying and the notion of wellbeing, which are different from western ideals of a good old age and death. See, for example: https://doi.org/10.1016/j.jaging.2021.100972

In order to better contextualize their study, the authors might also want to state their position in relation to the models of positive, active and successful ageing (and neoliberal Western assumptions of functionality and productivity), which focus on wellbeing and quality of life. 

Thank you for addressing these important issues in your study and best of luck with your future research.

Author Response

Dear Reviewers,

Thank you for taking the time to review and consider our manuscript. We appreciate the detailed and thoughtful feedback. Below in blue text we outline the changes made to address each of the reviewers’ comments.  

Reviewer 2

The authors set to discuss and analyse very timely and important issues related to poverty, loneliness and ageing. This qualitative study explores the challenges and strengths of poor older adults living in Hong Kong. The authors focus on a variety of aspects that include social isolation, mental health, self-esteem, activity levels, medical expenses, ageism, retirement, and housing insecurity, among others. They identify the strengths and discussion points for future gerontological research and practice, and give solutions on how to improve the quality of life for older adults living alone and in poverty. The authors provide an updated and relevant lit review on the issues presented in the paper and outline the most important risk factors associated with ageing in poverty and being alone, by taking into account both the micro and macro levels. The paper is well written and clearly presented. I have some recommendations, though, that could improve it: 

It would be interesting to explore in more detail the gender dimension and discuss whether women, who are aged by culture earlier than men and subject to the double standard of ageing, present different perceptions on ageing, self-esteem, and ageism, and social isolation. This is important as older women live longer and are more prone to old age loneliness and ageism, and, thus, may need different actions to improve their wellbeing and quality of life. More specific solutions for older women could be considered, if possible, by taking into account the particular cultural and social context (Hong Kong). 

Response:

Discussion point 4, on p. 12 was revised. Whereby, the focus is not only on discussing specific social participation activities for men, but also the need to explore how being an older adult female implicates age discrimination, self-esteem, self-efficacy, social isolation, and loneliness.

The following sentence was revised as follows:

“Fourth, the findings suggest gender differences to be explored further. First, there is a need to more deeply examine the barriers and facilitators to social participation for older men who live alone and in poverty.”

The following text was added to this discussion point:

Second, as the majority of the participants are female, this may suggest that the key challenges identified in the study (e.g., age discrimination, self-esteem, self-efficacy, social isolation and loneliness in late life) have a gendered dimension. There is empirical evidence that indicate older adult males are more likely to have higher levels of self-esteem (which is a predictor of life satisfaction) than their female counterparts [50]. Future research could explore how gender (specifically being female), age, and cultural context intersect to implicate discrimination, self-esteem, self-efficacy, social isolation and loneliness in late life.

The following was added to the reference list:

  1. Zhang, L.; Leung, J.-P. Moderating effects of gender and age on the relationship between self-esteem and life satisfaction in mainland Chinese. International Journal of Psychology, 2002, 37(2), 83–91.

Related to the findings on employment among older adults and suggested innovative solutions, the authors might consider life-long learning programmes and intergenerational relations. They already mention them, but could elaborate a bit more: see similar articles on intergenerational benefits in relation to old age.

Response:

Added the following text in discussion point three (on employment) on p 12:

Further, in some contexts, like Canada, such community-based lifelong learning programs for older adults at risk are publicly subsidized and have found to be an “important social determinant of health in an aging society” (p. 689) [48]. 

The following was added to the reference list:

  1. Narushima, M. More than nickels and dimes: the health benefits of a community-based lifelong learning programme for older adults. International Journal of Lifelong Education, 2008, 27(6), 673–692.

To improve wellbeing of older people, the authors could also reflect on more personal experiences of ageing, which are often ignored and undervalued in empirical research. They could mention storytelling or narratives on personal ageing experiences, which make older adults not the mere objects of study, but rather active subjects in research on health care and social policy services. 

Response:

The following text was added to the conclusion on p. 13:

“Such personal experiences of ageing are undervalued in empirical research but are important in transforming older adults from mere objects of study to active participants in research.”

Also, it would be interesting to see how sociocultural factors and contexts, in this case, living in Hong Kong, alter the perceptions of ageing, living, and dying and the notion of wellbeing, which are different from western ideals of a good old age and death. See, for example: https://doi.org/10.1016/j.jaging.2021.100972

In order to better contextualize their study, the authors might also want to state their position in relation to the models of positive, active and successful ageing (and neoliberal Western assumptions of functionality and productivity), which focus on wellbeing and quality of life. 

Response:

The following text has been added to the final discussion point on p. 13:

This finding (“I have enough”) also act as a point of reflection for us as researchers, who adopt a healthy ageing framework [54], which has emerged from largely Western perspectives that prioritize functionality and productivity. This theme is a reminder for us as researchers to step back from our dominant frame-work/lens and to recognize alternative perspectives of what constitutes health and well-being in late life.

The following reference is added to the reference list:

  1. World Health Organization. World report on ageing and health. Available online: http://www.who.int/ageing/publications/world-report-2015/en/ (accessed 20 November 2022)

Thank you for addressing these important issues in your study and best of luck with your future research.

Reviewer 3 Report

The study samples a common cross-section of people living alone and those living in poverty, but we do not know how these two characteristics correlate: are they independent, do they attract each other, i.e. do they occur more often together, or vice versa. It would also be important to know this in order to assess why the common cross-section is being studied, since many of the problems being studied are similar only for people living alone and only for the poor.

It would also be interesting to look at why people live alone. The study shows that it is not only those who do not have a family, but also many who do - e.g. have children. You would think that in such cases it would not be so much the role of the state and NGOs to help, but that of the family and the children.

Author Response

Thank you for taking the time to review and consider our manuscript. We appreciate the detailed and thoughtful feedback. Below in blue text we outline the changes made to address each of the reviewers’ comments.  

Reviewer 3

The study samples a common cross-section of people living alone and those living in poverty, but we do not know how these two characteristics correlate: are they independent, do they attract each other, i.e. do they occur more often together, or vice versa. It would also be important to know this in order to assess why the common cross-section is being studied, since many of the problems being studied are similar only for people living alone and only for the poor.

Response:

Added the following text in the subsection 1.2 “Older People Living Alone and in Poverty: The Intersection” to highlight what some studies in Hong Kong have found regarding the relationship between poverty and living alone:

In Hong Kong, studies have indicated that older adults living alone are not only more likely to be income-poor, but also asset-poor, socially excluded, and materially deprived. Also, the decrease in the number of wage earners in a household increase the risk of poverty [28-31].

Added in the reference list:

  1. Chan, L.; Chou, K. Immigration, living arrangement and the poverty risk of older

adults in Hong Kong. International Journal of Social Welfare, 2016, 25(3), 247-258.

  1. Chan, L.; Chou, K. A survey of asset poverty among older adults of Hong

Kong. Social Indicators Research. 2018, 138(2), 605-622.

  1. Chou, K. Social exclusion in old age: A validation study in Hong Kong. Aging &

Mental Health. 2018, 22(8), 1072-1079.

  1. Chou, K.; Lee, S. Superimpose material deprivation study on poverty old age

people in Hong Kong Study. Social Indicators Research. 2018, 139(3), 1015-1036.

It would also be interesting to look at why people live alone. The study shows that it is not only those who do not have a family, but also many who do - e.g. have children. You would think that in such cases it would not be so much the role of the state and NGOs to help, but that of the family and the children.

Response:

Added the following text in the last discussion point on p. 13:

Another theme related to this mindset is the “most of us like to live alone” theme, which indicates living arrangement preferences that are worthy of further exploration. Culturally, living with multiple generations is an ideal preference for Chinese older adults, however there is research to indicate a shift. For example, in Meng et al.’s [55] nationally representative sample of older people in Urban China, examining future living arrangement preferences of middle-aged and older people that living close to their children (e.g., in the same neighborhood) was the most popular preference, with living with adult children coming in second. 

Added the reference in the reference list:

  1. Meng, D.; Xu, G.; He, L.; Zhang, M.; Lin, D. What determines the preference for future living arrangements of mid-dle-aged and older people in urban China? PloS One, 2017, 12(7), e0180764–e0180764.